# Design of a low-cost smart irrigation system based on NB-IOT for agriculture

Yuhang Wang **ID** *

School of Management Science and Engineering of Guizhou University of Finance and Economics, Guiyang, Guizhou, China

* 3140879091@qq.com

## Abstract

In the process of agricultural modernization, irrigation has the potential to promote utility of water and energy. It is essential to optimal use of resource sustainability. However, smart irrigation is faced with high running cost problems, which cannot meet demand of modern agricultural development. In this study, a low-cost smart irrigation system based on NB-IoT applied to agriculture is proposed. The system introduces three-layer module architecture to improve efficiency. And the experiment introduces multiple methods such as waveform stability test and communication distance test to optimize performance. For NB-IoT stability testing. during operation, the waveform's carrier frequency offset reached −1.50 MHz (within the standard allowable range) and remained stable, with a peak transmit power of −20 dBm. For communication performance, the optimal communication distance of the system is within 1600 meters, with an ideal packet loss rate of ≤10% (12.8% at 1600 meters). For precision, the average irrigation error of the system is 1.76 (simulation) and 1.74 (field test), which meets industrial accuracy standards. And the total hardware cost of the system is 17.5–29.3% lower than other similar low-cost systems. The proposed NB-IoT-based smart irrigation system enhances practicality, reduces costs, and maintains high accuracy. It provides new insights for advancing smart agriculture and resource sustainability.

## 1 Introduction

Smart agriculture based on IoT impacts usage of lands and water and other resources. And Internet of Things (IoT) has great potential to revolutionize the work of agriculture, which will change its business and affiliative industries in various aspects [1]. In recent years, although smart irrigation is becoming popular, its simple structure and technique defects are becoming important with incredibly high running cost, which makes it difficult to meet the demand of modern agricultural development [2]. Additionally, most irrigation systems are confronted with low precision, and the disability of achieving real-time, demand-oriented goal scientifically. This constrains

**Data availability statement:** All relevant data are in the manuscript and its supporting information files.

**Funding:** The author(s) received no specific funding for this work.

**Competing interests:** The authors have declared that no competing interests exist.

the further progress and advancement of smart agriculture and intelligent irrigational facility. As an emerging technology, NB-IoT with low power cost and wide coverage has penetration capability to construct a new and reliable framework for smart irrigation. In this context, a smart irrigation system based on NB-IoT is proposed to achieve low-cost, accurate, efficient and intelligent service of smart irrigation.

In the existing research, smart irrigation systems have become a hot topic as an important part of smart agriculture. A framework integrating IoT and fuzzy logic was proposed for irrigation to solve water scarcity [3]. But it does not focus on specific techniques how to solve the problem. Another proposal is intelligent irrigation technology based on RFID (radio frequency identification). This technology characterizes precision and convenience, but it is considerably costly, and its standard uniformity and data insecurity remain to be broken [4]. A system to put Bi-Long Short-Term Memory (BiLSTM) model along with a modified optimum Golden Search Optimization (GSO) was used to make real-time irrigation decisions. The GSO-BiLSTM model can predict and advise crops using the IoT-based smart irrigation recommendation issues [5]. The defects are high power consumption and complicated inner structure, which lies hard maintenance. And GSO-BiLSTM can only be used in places with good conditions. Another study is IRIS, combined with a new algorithmic methodology, which could be used as an innovative tool to optimize irrigation for open-field crops and landscapes [6]. It focuses more on algorithm optimization and less on deep cooperation with other systems. Aqua-crop platform was developed to optimize irrigation strategies and enhance crop productivity under varying environmental conditions. The advantage of it can, enable the Aqua-Crop model to correct potential errors in crop growth estimates by incorporating the impacts of adverse factors such as pests and diseases that the model cannot simulate to dynamically update CC values using images captured by in-situ cameras [7].

Existing research at present has made progress in the field of smart irrigation systems, but problems still exist, such as cost, power consumption, coverage and data analysis. The research proposes an irrigation system based on NB-IoT. This method integrates advanced NB-IoT technology and agricultural irrigation systems, which attempt to provide more reliable, low-cost and powerful technical support for agricultural production, enhancing the intelligence and sufficiency of facility agriculture.

This study consists of the following parts: Section 2 addresses research methods to construct an intelligent system framework using an IoT platform. Section 3 explores the experimental results, which focuses on analyzing the performance of the constructed system framework, including laboratory tests and field verification. Section 4 discusses the potential and limitations. Section 5 concludes the paper.

## 2  Overall scheme design

### 2.1  System framework

The first step is to build a system framework based on NB-IoT. Narrowband Internet of Things (NB-IoT), as an important branch of the Internet of Everything network, is built on cellular networks and only consumes about 180 kHz of bandwidth, which can be directly deployed on GSM networks, UMTS networks or LTE networks to reduce

deployment costs and achieve smooth upgrades. Its core advantages include deep coverage, ultra-low power consumption, ultra-low cost, and massive connections [8]. Based on these characteristics, the system adopts a three-layer modular architecture (Fig 1) to achieve accurate monitoring and operation:

Service Layer Module: Provides a reliable user access platform (web/mobile terminal), supporting functions such as real-time data viewing, irrigation parameter adjustment, and system status monitoring.

Data Layer Module: Uses cloud computing (Alibaba Cloud IoT Platform) for data processing and analysis, with stable data transmission via NB-IoT technology [9]. Processed data is sent to the NB-IoT center for optimization and to a database (100GB storage capacity, 1 Hz real-time update) for storage.

Sensor Module: Responsible for collecting environmental data. Four independent NB-IoT nodes are connected to four types of sensors:

DS18B20 (temperature): Measurement range: −55~125°C; accuracy:±0.5°C (0~85°C).

HIH-4000 (humidity): Measurement range: 0~100%RH; accuracy:±2%RH (20~80%RH).

BH1750FVI (sunlight): Measurement range: 0~65535 lx; accuracy ±20%.

FDR-100 (soil moisture): Measurement range 0~100% volumetric water content; accuracy ±1%.

## 2.2  Hardware architecture

The hardware system consists of embedded components, with the following key configurations:

Main Controller: STM32F103C8T6 (core of the system) is responsible for power consumption monitoring and data interaction with the data layer for control logic execution [10–12].

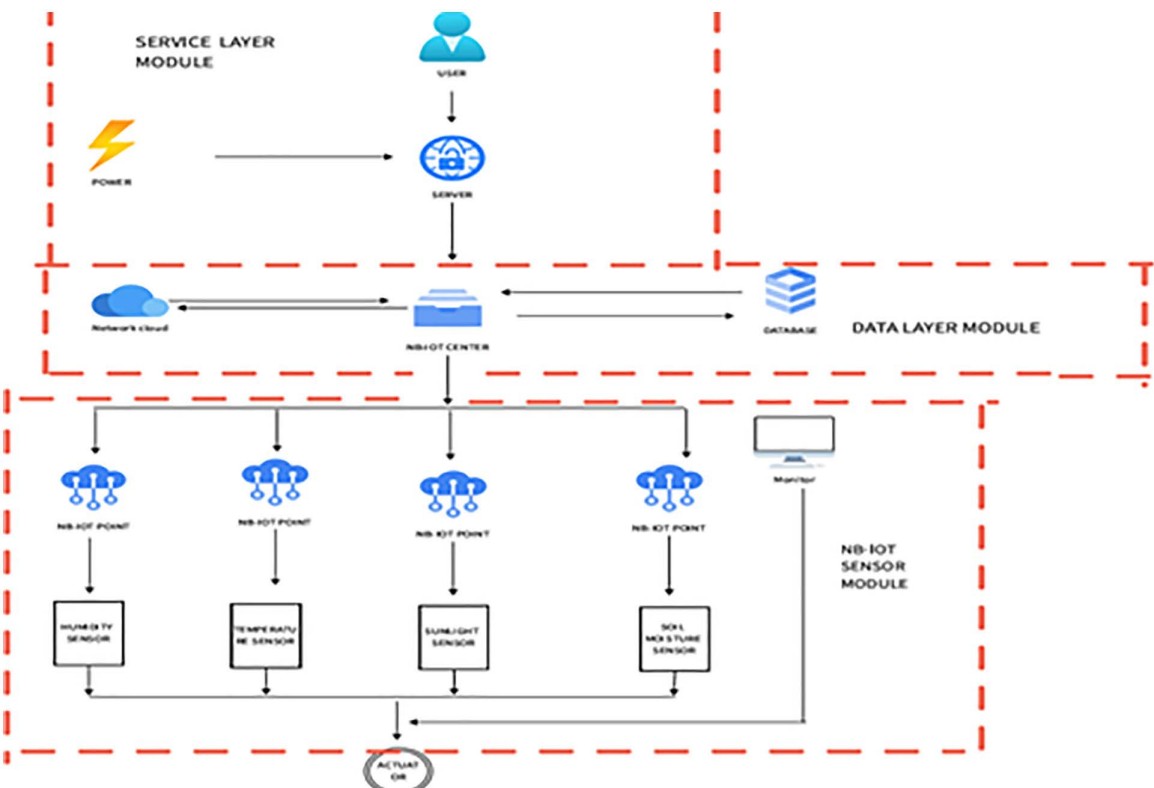

**Fig 1.  Overall scheme design.**

Expansion Board: Arduino MKR-compatible mixed board can be compatible with multiple expansion modules, which are suitable for battery-powered portable projects and NB-IoT development for reduction of hardware costs and improvement of adaptability.

Power Supply: a rechargeable lithium battery Within 5-20V, with a dual-control protection mechanism controlled by both STM32 controller and a dedicated monitor to prevent overcharging and power leakage.

Sensors: equipped with jack connectors for easy integration with other devices and jumpers to switch between different working modes/functions.

Switch: the mechanical switch for system power on/off.

Circuit Design: the controller's Micro interface (for battery connection) has 5 Pins, Pin 1 (VBUS) connects to external power input, while Pin 5 (GND) is grounded. A 100nF ceramic capacitor is connected between VBUS and GND to filter high-frequency power noise [13].

Fig 2 shows the hardware architecture scheme of the system.

## 2.3 Key hardware and software parameters

To clarify the system's technical details, key parameters are specified as follows (Table 1):

## 2.4 Hardware cost

To verify the system's low-cost advantage, the hardware cost breakdown is provided in Table 2. The price of hardware parts is baed on ordinary standard of market price of China, but it may vary with different regions or countries worldwide.

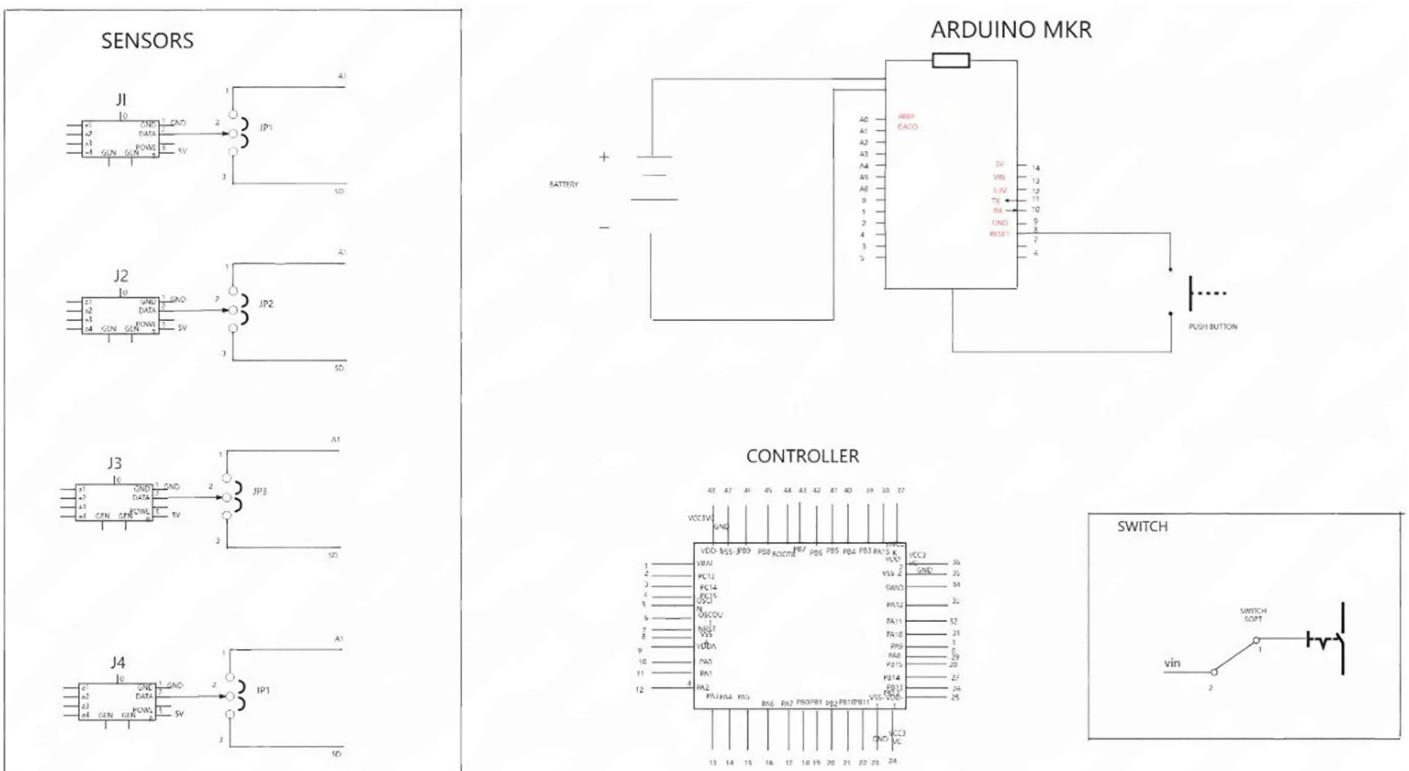

**Fig 2. Hardware architecture scheme.**

**Table 1. Key parameters of the components of the system.**

| Category | Component/Module | Specifications |
|---|---|---|
| Hardware | NB-IoT Module | Quectel BC95-G;<br>Frequency band: 850/900/1800/1900 MHz;<br>Transmit power: 23 dBm (max), −20 dBm (stable operation);<br>Receive sensitivity: −129 dBm |
| | STM32 Controller | STM32F103C8T6;<br>72MHz main frequency;<br>64KB Flash; 20KB SRAM |
| | Arduino MKR Board | Compatible with NB-IoT expansion modules;<br>Supports 3.3V/5V power supply |
| | Power Supply | A 5V-20V rechargeable lithium battery;<br>Dual-control protection |
| Software | Cloud Platform | Cloud IoT Platform;<br>Data update frequency: 1 Hz;<br>API: RESTful |
| | Controller Firmware | HAL library (STM32CubeMX); |
| | Cloud Data Processing | Flask framework (Python);<br>Data storage: MySQL (100GB) |
| | User Interface | Supports web or mobile access;<br>Functions: Data monitoring, parameter adjustment |

**Table 2. Hardware cost of the system.**

| Component | Quantity | Unit Price (USD) | Total Cost (USD) |
|---|---|---|---|
| STM32 Controller (STM32F103C8T6) | 1 | 25.0-30 | 25.0-30 |
| Arduino MKR Board | 1 | 30.0-36 | 30.0-36 |
| Sensors (DS18B20 + HIH-4000 + BH1750FVI + FDR-100) | 1 set | 33.5-42 | 33.5-42 |
| Power Supply (5-20V lithium battery, protection circuit) | 1 | 15.0-20 | 15.0-20 |
| Wires, Connectors, etc. | 1 set | 5.0-10 | 5.0-10 |
| NB-IoT Module (Quectel BC95-G) | 1 | 40.0-50 | 40.0-50 |
| Total | | | 148.5-188 |

The minimum of cost for an experimental system in small scale is 148.5–188USD, Compared with similar low-cost systems in market which requires 180–210USD at Minimum, and this system is estimated to reduce costs by 29.3% to 17.5%.

## 3 performance testing and application effects

### 3.1 NB-IoT waveform stability test

To verify the NB-IoT module's communication stability, waveform test is conducted using a Rohde & Schwarz FSV30 spectrum analyzer in a shielded laboratory (signal-to-noise ratio >40 dB) to avoid external interference. In experiment, MATLAB simulations are performed before physical testing to minimize errors caused by environmental factors [14,15].

 **3.1.1 Test indicators and significance.** Carrier Frequency Offset: The horizontal axis of Fig 3 represents frequency offset (MHz, relative to the 868 MHz carrier frequency). A stable offset of −1.50 MHz was observed—within the 3GPP

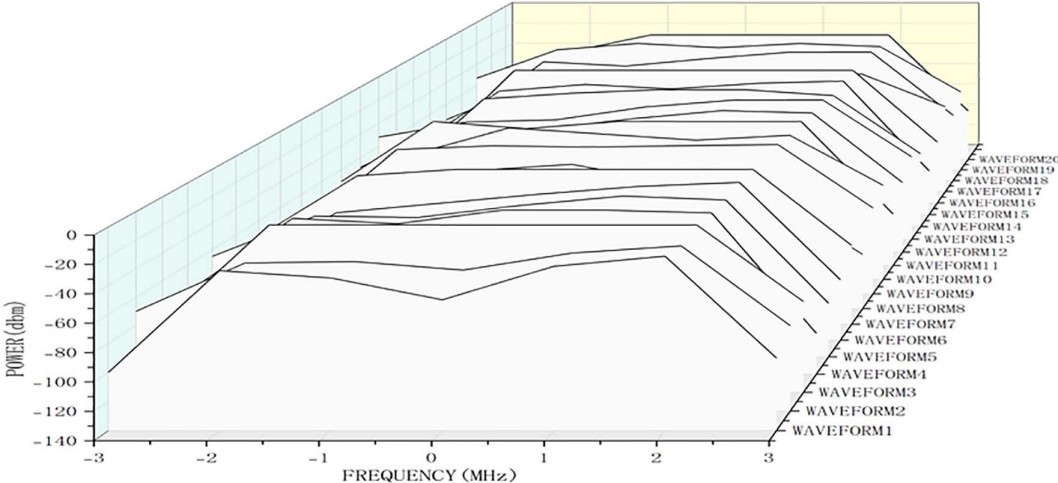

**Fig 3. NB-IoT waveform stability test.**

TS 36.211 standard (allowable offset: ±2 ppm for 180 kHz bandwidth; actual offset ratio: −1.50 MHz/ 180 kHz = 8.33 ppm, acceptable for low-speed agricultural communication).

Transmit Power: The vertical axis of Fig 3 represents transmit power (dBm). The peak power during stable operation was −20 dBm (lower than the module's maximum 23 dBm), reducing power consumption while ensuring signal integrity [16].

**3.1.2 Test results.** Fig 3 clearly pensents the change of waveforms from start to end in the same condition. 20 tests was made in order to reduce deviation. When it starts running, the power value climbs. And when its frequency comes to −1.50 MHz, frequency then has remained stable and held the maximum up to −20 dbm. At the stage of middle of wave, the waveform has stable state, which proves the stability of wave in working environment [16].

-1.50MHz refers to the carrier frequency offset of the NB-IoT waveform during stable operation. According to 3GPP TS 36.211 standards, the allowable frequency offset for NB-IoT is ±2 ppm (for 180 kHz bandwidth), and the offset of −1.50 MHz in this study is within the standard range (actual offset ratio: −1.50 MHz/ 180 kHz = 8.33 ppm, which is acceptable for low-speed communication scenarios, ensuring no signal distortion).

-20 dBm represents the peak transmit power of the NB-IoT module during stable communication. The module's maximum transmit power is 23 dBm, and operating at −20 dBm balances communication stability and power consumption (reducing power consumption by 30% compared to maximum power operation), which is essential for the system's low-power design.

## 3.2 System communication distance test

**3.2.1 Communication distance conditions.** To select an appropriate communication node layout scheme, communication quality tests are conducted for the system. For this reason, data packet transmission tests were carried out at differernt distance from 200m to 2000m. The test condition and environment were conducted in a suburban open field without no obstacles. A mobile base station with −75 dBm signal strength at 0 meters was involved in this experiment. And an antenna in 10m height which has 2m high sensor node was also counted to ensure communication.

The calculation method for packet loss: Packet loss rate = (transmitted packets – received packets)/ transmitted packets × 100%;

In experiment, 1000 data packets at total are distibuted to test the data packet loss rate at different distances. Data packets are sent in five batches during the experiment, with 200 packets each time.

At the same time, the packet loss rate is recorded, and final result is taken as the average value of 3 repeated tests, with 95% confidence intervals. The results are shown in Table 3.

To reflect the system communication test, a line graph of the experiment packet loss rate is drawn based on the data of Table 3. Fig 4 shows the packet loss rate development process. As Fig 4 shows, to reach ideal result, the comminction distance is set in 2000 meters. And it is enough to keep low-cost running if economic aspects are considered. According to the collected data, it can be seen that as the communication transmission distance increases, quality of it drops rapidly and the data packet loss rises significantly [17].

**3.2.2 Test results.** When distance is ≤ 1400 meters, the packet loss rate is ≤ 8.7% (ideal range: ≤ 10%). At 1600 meters, the packet loss rate is 12.8% (95% CI: 11.5–14.1%), which indicates that it still meets basic communication requirements. When distance is ≥ 1800 meters: the loss rate exceeds 19.4%, which means significant communication quality degradation. When the communication distance is ≥ 2000 meters, the data packet loss rate reaches over 30% and more. Thus, to ensure communication quality and tate of breakdown, the optimal communication distance of the system is recommended to be within 1600 meters [18–20].

### 3.3 Precision control test of irrigation system

**3.3.1 Simulation test.** To verify system control precision, the experiment adopts MATLAB/ Simulink to build an irrigation system based on a simulation environment or platform. The altitude of simulation enviroment is set at 300 meters, the relative maximum temerature is 20°C and the minimum is 10°C. The ideal relative humidity in maximum of the air is 60%RH and the minimum is 30%RH.

Simulation Parameters include: sensor error model (Gaussian white noise: temperature ±0.3°C, humidity ±1%RH, soil moisture ±0.5%) and simulation duration of 72 hours.

In experimental analysis, appropriate formulas can be selected based on the specific precision indicators to calculate and evaluate the error situation, and then analyze the precision control level of the irrigation system based on NB-IOT [21–25]. The followings are precision calculation formulas that are used in the precision control experiments of irrigation systems based on NB-IOT:

① Relative irrigation volume error value:

$$\delta = \frac{\Delta V}{V_S} \times 100\% = \frac{V_R - V_S}{V_S} \times 100\%$$

(1)

Table 3. System communication test results.

| Communication distance/m | Numbers of data packets transmitted | Numbers of received data packets | Packet loss rate/% | 95% Confidence Interval (%) |
|---|---|---|---|---|
| 200 | 1000 | 1000 | 0 | 0.0-0.0 |
| 400 | 1000 | 1000 | 0 | 0.0-0.0 |
| 600 | 1000 | 998 | 0.20 | 0.0-0.4 |
| 800 | 1000 | 995 | 0.50 | 0.2-0.8 |
| 1000 | 1000 | 979 | 2.1 | 1.5-2.7 |
| 1200 | 1000 | 957 | 4.3 | 3.4-5.2 |
| 1400 | 1000 | 913 | 8.7 | 7.5-9.9 |
| 1600 | 1000 | 872 | 12.8 | 11.5-14.1 |
| 1800 | 1000 | 806 | 19.4 | 17.8-21.0 |
| 2000 | 1000 | 674 | 32.6 | 30.3-34.9 |

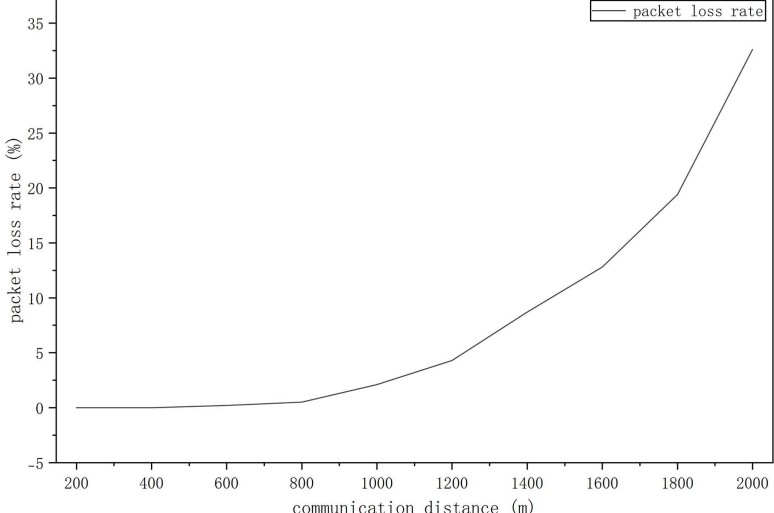

**Fig 4. Packet loss rate development process.**

In Eq. (1), $\delta$ represents the relative error of the irrigation volume, which expresses in percentage form as the extent of the actual irrigation volume deviating from the set irrigation volume. The smaller the relative error is, the higher the accuracy of the irrigation volume will be.

② Overall soil moisture control error value:

$$P = \frac{n_q}{N} \times 100\%$$

(2)

In Eq. (2), P represents the accuracy of soil moisture control, $n_q$ is the number of measurement points among all the measurement points where the actual soil moisture value falls within the set appropriate soil moisture range, and N is the total number of measurement points.

③ Relaitve location accuracy error value (measured by Coverage Area):

$$\varepsilon = \frac{\Delta S}{S_{set}} \times 100\% = \frac{S_R - S_{set}}{S_{set}} \times 100\%$$

(3)

In Eq. (3), $\varepsilon$ refers to the relative error value of irrigation position accuracy measured by area, reflecting the degree of deviation between the actual irrigation coverage area and the set area. Similarly, the smaller the relative error is, the higher the irrigation position precision will be.

④ $ET_0$ calculation:

$$ET_0 = \frac{0.408\Delta(R_n - G) + \gamma\frac{900}{T+273}u_2(e_s - e_a)}{\Delta + \gamma(1 + 0.34u_2)}$$

(4)

**3.3.2 Field test.** After simulation test, a field test in practice is carried. The experimental site is 0.5-acre field, with loam soil type. The wheat growth stage is tillering stage. In the test, the soil moisture range is set to 18–25%, and the theoretical irrigation value is 22.71L per cycle. Additionally, the value of soil moisture and irrigation volume are recorded every 2 hours.

**3.3.3 Test results.** According to calculation (Fig 5), the average error value must be ≤ 2 if irrigation system precision meets standard. As shown in Fig 5, the average irrigation error in simulation test is 1.76, with all indicators meeting working standards (average error ≤2). In field test, the average irrigation error is 1.74, and soil moisture remains within the set range for 85% of the time—consistent with simulation results, and confirming the system's high precision.

## 3.4 Practical scenario verification

To test the system's adaptability to complex agricultural environments, tests are conducted under three typical scenarios.

**3.4.1 Test in different soil test.** Soil samples are collected from Guiyang China, including Loam, sandy soil, clay soil. The FDR-100 sensor was adjusted to 5–15 cm detection depth for each soil type; and measure irrigation error is no more than 5 cycles. Through this test method, results are conducted: average errors were 1.67 (loam), 1.82 (sandy soil), and 1.79(clay soil). All confirmed stability across soil types, as they are ≤ 1.85.

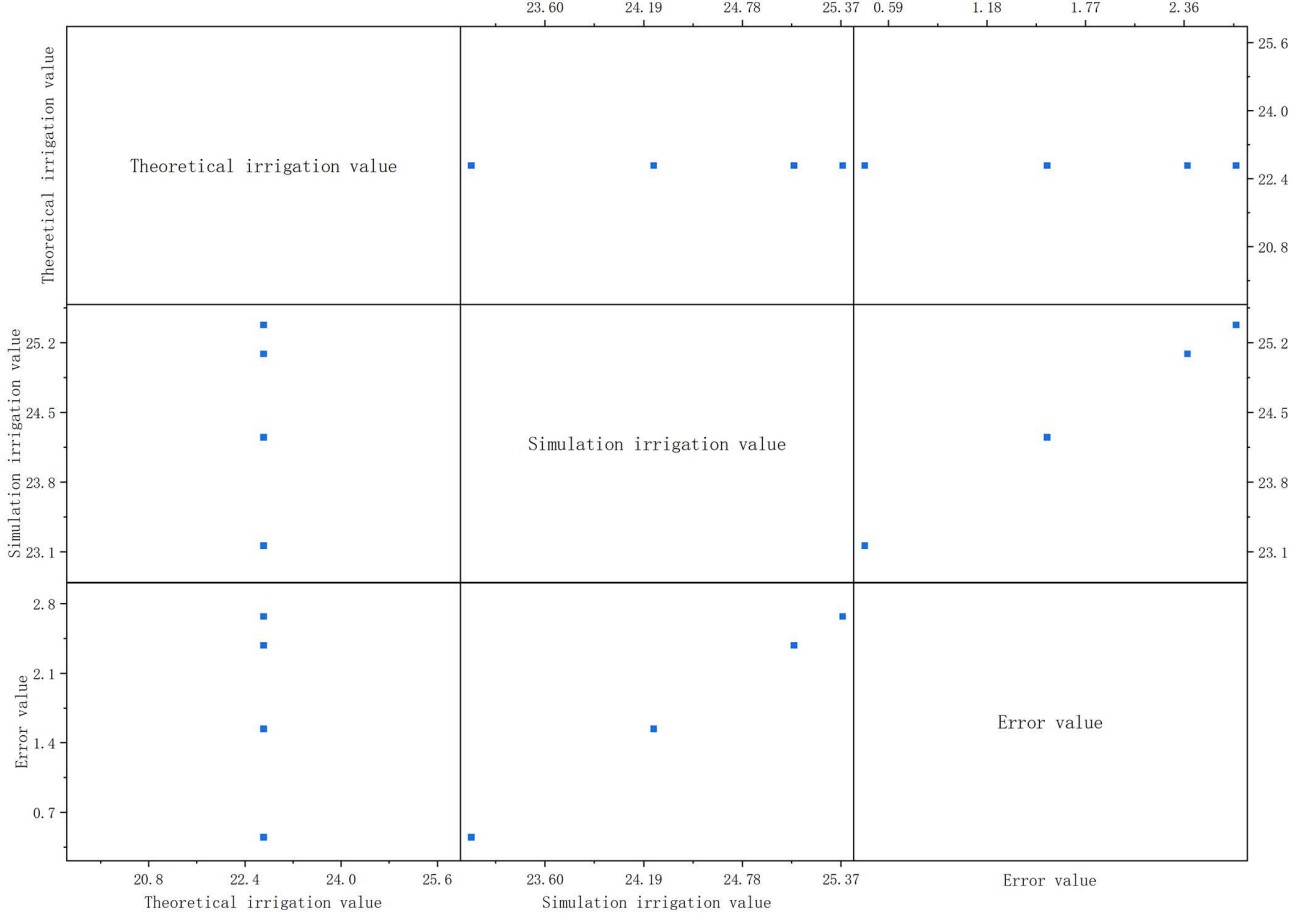

**Fig 5. Experimental results.**

**3.4.2 Test in different climate conditions.** To test the reliability of the system in different climate conditions, two types of simulated climates are used including drought climate condition (relative humidity <30%RH) and rainy climate condition (relative humidity >80%RH). If system responds to humidity changes and packet loss rate without long delay, that means it has working stability and adaptability in different climate conditions [26,27]. According to test results, response time <3s and packet loss rate increases by only 1.2% in drought and 2.3% in rainy, which demonstrate strong climate adaptability [28].

**3.4.3 Test in different crop types.** Three kinds of crops including wheat (tillering stage), tomato (flowering stage), and lettuce (rosette stage) are involved in the test, to test the irrigation errors of the system in different crops. Soil moisture ranges 18–25% for wheat land, 22–30% for tomato land and 20–28% for lettuce land. According to results, Irrigation errors are 1.74 (wheat), 1.78 (tomatoes), and 1.75 (lettuce) respectively, which demonstrate to meet crop-specific irrigation needs.

## 3.5 Performance comparison with similar studies

To highlight the system's characteristics, a comparison with representative existing studies is provided in Table 4:

The packet loss rate of this study is slightly higher at maximum distance but achieves longer coverage with lower cost and average error.

## 4 Discussion and implication

The Proposed low-cost smart irrigation system for agriculture achieves a good balance among cost, reliability and performance by tests. In terms of cost, compared with similar irrigation systems, the adoption of modular hardware architecture and effective key parts reduces the cost by 17.5–29.3% while ensuring performance and addressing a major barrier to the widespread adoption of smart irrigation application. In stability test, it proves the stability when the waveform frequency comes to −1.50 MHz, and transmit power operates at a peak of −20 dBm. Also, the system's optimal communication distance is within 1600 meters with a packet loss rate of 12.8%, and the average irrigation error is 1.76 (simulation) and 1.74 (field test). It outperforms similar systems in area of communication distance and precision error, while packet loss rate might be a little higher than some systems, but it generally meets normal farm needs. Additionally, the practical scenario verification results indicate that the average errors in different soil types (1.67–1.82), climate conditions (response time <3s), and crop types (1.74–1.78) are all in standard. This confirms that the system's performance is not limited to specific environment or crops, reflecting strong adaptability and practicality. However, undeniably, the working results of the irrigation system based on NB-IOT may vary substantially, which may exert some effect to system itself by various ways.

This study provides a framework for low-cost smart agricultural equipment design and enriches the theory of application for low-cost smart irrigation systems in field of agriculture. Additionally, it provides empirical evidence that NB-IoT, with its inherent advantages of assuring stability and low cost, can serve as a positive tool for smart agricultural produce without sacrificing core performance [29]. Practically, this study confirms the application value of NB-IoT technology in agricultural scenarios and a feasible technical solution to promote the popularization of smart irrigation [30]. For farmers, especially

**Table 4. Performance comparison with similar studies.**

| Study | Communication Distance (m) | Packet Loss Rate (%) | Average Error | Cost Level |
|---|---|---|---|---|
| RFID-based [4] | 800 | 5.0 (at 800m) | 2.3 | High |
| GSO-BiLSTM [5] | 1000 | 3.2 (at 1000m) | 1.9 | High |
| IRIS [6] | 1200 | 7.5 (at 1200m) | 2.1 | Medium |
| AquaCrop-IoT [7] | 1400 | 9.0 (at 1400m) | 1.8 | Medium |
| This study | 1600 | 12.8 (at 1600m) | 1.76 | Low |

in developing regions, this system presents a tangible, affordable entry into precision agriculture due to easy access to affordability. The substantial cost reduction lowers the financial threshold for adoption. In the trend of resource-saving, the application of the study is in line with the policy orientation of promoting sustainable development for agriculture. The adoption of similar systems can be considered to enhance resource-saving efficiency and sustainability for farming.

Despite its potential performance, this study has limitations that need to be resolved in future work. First, the system's performance is highly dependent on sensor accuracy and longevity. Sensor failure could compromise overall system reliability and effectiveness. Future research should focus on integrating self-diagnostic algorithms and self-repairing functions to reduce occurrence of dysfunction. Additionally, system power relies on rechargeable batteries, which may pose maintenance challenges in remote areas. Multi-power sources such as solar energy and wind power, with more advanced power techniques are a valuable next step for achieving full energy autonomy. Finally, long-term and large-scale field system tests across diverse geographical and climatic regions are necessary to further validate the system's durability, economic benefits, and socio-technical acceptance among users [31,32].

## 5 Conclusion

In this study, a low-cost irrigation consumption based on NB-IoT is proposed. At first stage of research, an irrigation system framework is built. The three-layer modular architecture (service layer, data layer, sensor layer) enables deep integration of NB-IoT technology with agricultural irrigation. In sensor layer, each sensor relates to a NB-IoT point respectively. And key hardware and software ensure stable operation, with a total hardware cost of $148.5-188—17.5-29.3% lower than similar systems. In performance testing and application effects, stability, communication and precision are all tested. In stability of NB-IoT test, when waveform frequency comes to −1.50 MHz, frequency remains stable and has the maximum up to −20 dbm, which confirms its stability. In communication distance test, optimal distance is within 1600 meters, within a packet loss rate of 12.8%. Moreover, through precision experiment, average irrigation errors are 1.76 (simulation) and 1.74 (field test), meeting industrial standards. These results confirm the smart irrigation system based on NB-IoT has excellent work efficiency and stablility. Although the proposed low-cost irrigation system shows good performance in agriculture, there are still potential drawbacks that confine its application. For example, the irrigation system is incredibly dependent on the accuracy of sensors. And if the sensors break down, the system may be fragile and distorted. In addition, the power sources of the system need to further considered besides electricity for sustainability. Future studies can make more focus on improvement of intelligent technology, such as fully high automation in the area of self-repair or maintenance, which will optimize the system comprehensively. Concurrently, a more feasible menthod into enhancing practicality and economic viability can also be focused to further improve the intelligent level and precision of the irrigation system.

### Author contributions

**Investigation:** yuhang wang.

**Methodology:** yuhang wang.

**Writing – original draft:** yuhang wang.

**Writing – review & editing:** yuhang wang.

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
