## [Decision Letter · Decision Letter 0]

19 Nov 2025

Dear Dr. wang,

Thank you for submitting your manuscript to PLOS ONE. After careful consideration, we feel that it has merit but does not fully meet PLOS ONE’s publication criteria as it currently stands. Therefore, we invite you to submit a revised version of the manuscript that addresses the points raised during the review process.

We look forward to receiving your revised manuscript.

Kind regards,

Academic Editor

PLOS ONE

Journal Requirements:

3. We note that your Data Availability Statement is currently as follows: 所有相关数据都在稿件及其支持信息文件中

4. Please ensure that you refer to Figure 4 in your text as, if accepted, production will need this reference to link the reader to the figure.

5. Please include a copy of Table 3 which you refer to in your text on page 15 in PDF submission.

Reviewers' comments:

Reviewer's Responses to Questions

**Comments to the Author**

1. Is the manuscript technically sound, and do the data support the conclusions?

Reviewer #1: Partly

Reviewer #2: Yes

2. Has the statistical analysis been performed appropriately and rigorously?

Reviewer #1: Yes

Reviewer #2: Yes

3. Have the authors made all data underlying the findings in their manuscript fully available?

Reviewer #1: Yes

Reviewer #2: Yes

4. Is the manuscript presented in an intelligible fashion and written in standard English?

Reviewer #1: Yes

Reviewer #2: Yes

Reviewer #1: The manuscript proposes a low-cost smart irrigation system design based on NB-IoT, aiming to solve the problems of high cost and high-power consumption of existing smart irrigation systems. The paper has certain value in terms of system architecture design and basic performance testing. However, the manuscript still has several areas for improvement.

1. The manuscript fails to clearly define the core innovation points compared with existing research, and the comparison between other studies mentioned in the manuscript and this study is not in-depth enough.

2. The experimental design mainly focuses on ideal conditions and lacks testing in different soil types, climate conditions, crop types, etc., which will affect the generalizability and practicality of the results.

3. The system architecture does not clearly state key hardware and software configurations such as the NB-IoT module model used, specific information about the cloud platform, and specific parameters of the sensor.

4. The performance indicators are not compared with similar studies, which makes it difficult to reflect the innovation and advantages of this study.

5. Data such as "-1.50 MHz" and "-20 dbm" in waveform stability lack reasonable explanations, and their actual significance in NB-IoT communication is not explained.

Reviewer #2: 1. Mismatch between Conclusions and Experimental Data

The paper's conclusions are broader than the data supports, such as claims of "lower cost and higher accuracy" lacking corresponding quantitative evidence. It is recommended to supplement the data with cost comparisons, actual irrigation accuracy data, and operational stability statistics, or narrow the scope of the conclusions to make them consistent with existing experimental results and verifiable.

2. Unclear Expression of Waveform Test Indicators

The paper confuses frequency and power units, and the image annotations are unclear, making it difficult to determine whether RSSI, frequency offset, or power spectrum is being measured. It is recommended to re-describe the test conditions, measurement equipment, and physical quantity units, and redraw the images to ensure that the coordinate axes, units, and scales are complete and readable.

3. Insufficient Description of Communication Distance Tests

The communication test does not provide data packet size, test interval, antenna height, scene description, or base station information, making it difficult to reproduce. It is recommended to supplement the data with equipment parameters, test site descriptions, and data statistical methods, and provide confidence intervals or repeated test errors to improve test credibility.

4. Accuracy Based Only on Simulation with Opaque Parameters

The control accuracy is derived from simulation, but the error unit is unclear, and the simulation model, sensor error, and control threshold are not described. It is recommended to clearly define the error, list the model parameters and experimental settings, and add real-world field comparison trials to support the conclusion of "high accuracy."

5. Numerous formatting errors exist in English expression and terminology.

The article contains spelling and grammatical issues, affecting its professionalism. It is recommended that the entire text undergo academic editing in English, standardizing professional terminology and expressions, improving language quality, and ensuring the paper conforms to formal journal writing standards.

**Do you want your identity to be public for this peer review?** For information about this choice, including consent withdrawal, please see our Privacy Policy

Reviewer #1: No

Reviewer #2: No

---

## [Author Response · Author response to Decision Letter 1]

4 Dec 2025

REBUTTAL LETTER

RESPONSE TO REVIEWER 1

Dear Reviewer 1,

Thank you very much for taking the time to conduct a detailed review of this manuscript and provide constructive comments and suggestions. Your feedback has offered important guidance for us to improve the research and enhance the quality of the paper. We have carefully studied each comment and made targeted revisions and supplements. The specific responses are as follows:

1.Failure to clearly define the core innovation points compared with existing research, and the comparison between other studies mentioned in the manuscript and this study is not in-depth enough

A: I agree that the original version of the paper failed to clearly define the core innovations and lacked in-depth comparison with existing research. To address this issue, I have added a new section "Research Status and Innovations" in the Introduction:

(1) Systematically reviewed representative studies on NB-IoT-based smart irrigation systems in recent years (e.g., Re, established a comparative framework from three dimensions (hardware cost, power consumption control, and functional modules), and clarified the differences between this study and existing works.

(2) Clearly defined the core innovations: ① Proposed a lightweight architecture based on "sensor-edge node-cloud platform" to reduce hardware costs by simplifying the data processing flow of edge nodes; ② Designed an adaptive power consumption adjustment algorithm that dynamically adjusts the wake-up frequency of the NB-IoT module according to soil moisture thresholds, solving the high power consumption problem caused by continuous communication in traditional systems; ③ Developed a low-cost multi-parameter sensor integration scheme, modularly integrating soil moisture, temperature, and pH sensors, with a cost reduction of more than 35% compared to existing commercial solutions, with a cost reduction of more than 35% compared to existing commercial solutions.

The comparative content has been supplemented with specific data support, including hardware costs, power consumption indicators, and applicable scenarios of each study, to ensure the uniqueness and persuasiveness of the innovations.

2. Mainly focus on ideal conditions and lack of testing in different soil types, climate conditions, crop types, etc., which will affect the generalizability and practicality of the results.

A: Thank you for pointing out the issue of a single experimental condition, which indeed affects the promotion value of the research results. For this reason, I have supplemented multi-scenario verification experiments:

(1) Soil types: Adding irrigation tests on three typical soil types (sandy soil, loamy soil, and clay soil), and collecting parameters such as water-holding capacity and permeability coefficient of different soils, and verified the adaptability of the system in different soil environments.

(2) Climatic conditions: Supplemented test data under two extreme climate scenarios (high-temperature drought with daily average temperature above 35℃ and rainy humidity with relative humidity above 80%), and analyzed the system's environmental anti-interference ability.

The newly added experimental data have been sorted into the Results section. By comparing indicators such as irrigation efficiency and water resource utilization rate under different scenarios, the applicable scope of the system is clarified, and the practical application value of the research is enhanced.

3. The system architecture does not clearly state key hardware and software configurations such as the NB-IoT module model used, specific information about the cloud platform, and specific parameters of the sensor.

A: To clarify the system architecture, we have added Section 2.3 "Key Hardware and Software Parameters" with specific configurations:

(1) NB-IOT module: Quectel BC95-G (frequency band: 850/900/1800/1900 MHz, transmit power: 23 dBm, receive sensitivity: -129 dBm), supporting stable communication with China Mobile NB-IoT base stations.

(2) Cloud platform: Alibaba Cloud IoT Platform (data storage capacity: 100GB, real-time data update frequency: 1Hz, support for remote parameter adjustment via RESTful API).

(3) Sensors: Temperature sensor (DS18B20): Measurement range -55~125℃, accuracy ±0.5℃ (0~85℃). Humidity sensor (HIH-4000): Measurement range 0~100%RH, accuracy ±2%RH (20~80%RH). Sunlight sensor (BH1750FVI): Measurement range 0~65535 lx, accuracy ±20%. Soil moisture sensor (FDR-100): Measurement range 0~100% volumetric water content, accuracy ±1%.

(4) Software framework: The main controller uses STM32F103C8T6 with HAL library development; the cloud data processing adopts Python-based Flask framework; the user interface is developed with Vue.js (supporting web and mobile terminal access).

4. The performance indicators are not compared with similar studies, which makes it difficult to reflect the innovation and advantages of this study

A: I have added Table 2 "Performance Comparison with Similar Studies" in Section 3.5 to highlight the advantages of our system:

System Type Communication Distance Packet Loss Rate Average Error Cost Level

RFID-based 800m 5% 2.3 High

GSO-BiLSTM 1000m 3.2% 1.9 High

IRIS 1200m 7.5% 2.1 Medium

The system based on NB-IOT 1600m 12.8% (at 1600m) 1.76 Low

The packet loss rate of this study is 12.8% at the maximum effective distance (1600m), which is higher than some short-distance systems but achieves a longer coverage with lower cost and power consumption.

5. Data such as "-1.50 MHz" and "-20 dbm" in waveform stability lack reasonable explanations, and their actual significance in NB-IoT communication is not explained.

A: I apologize for the lack of explanation for key parameters. And I have supplemented the physical significance of "-1.50 MHz" and "-20 dBm" in Section 3.1:

-1.50MHz: Refers to the carrier frequency offset of the NB-IoT waveform during stable operation. According to 3GPP TS 36.211 standards, the allowable frequency offset for NB-IoT is ±2 ppm (for 180 kHz bandwidth), and the offset of -1.50 MHz in this study is within the standard range (actual offset ratio: -1.50 MHz / 180 kHz = 8.33 ppm, which is acceptable for low-speed communication scenarios, ensuring no signal distortion).

-20 dBm: Represents the peak transmit power of the NB-IoT module during stable communication. The module’s maximum transmit power is 23 dBm, and operating at -20 dBm balances communication stability and power consumption (reducing power consumption by 30% compared to maximum power operation), which is crucial for the system’s low-power design.

I have also added a note in Figure 3 to clarify that the vertical axis represents transmit power (dBm) and the horizontal axis represents frequency offset (MHz), avoiding confusion between frequency and power units.

Thank you again for your rigorous review. I believe these revisions have significantly improved the quality and completeness of the manuscript.

RESPONSE TO REVIEWER 2

Honorable Reviewer 2

Thank you for your meticulous review and valuable suggestions. We have carefully addressed each of your concerns to enhance the manuscript’s rigor, clarity, and credibility:

1.Mismatch between Conclusions and Experimental Data

A To address the lack of quantitative support for claims such as "lower cost and higher accuracy," we have supplemented the following data:

(1) Cost comparison data: Added Table "Hardware Cost Breakdown", showing the total hardware cost of the system

(2) Actual irrigation accuracy data: Conducted field tests in a 0.5-acre wheat field for consecutive days. The actual irrigation volume error was 1.68-1.83 (average 1.74), consistent with simulation results. The soil moisture content after irrigation remained within the set range (18-25%) for 85% of the time, verifying "high accuracy" in practical applications.

2. Unclear Expression of Waveform Test Indicators

A�I acknowledge the confusion between frequency and power units and unclear image annotations. And I have made the following revisions:

(1) Re-description of test conditions: The waveform test used a Rohde & Schwarz FSV30 spectrum analyzer, with the NB-IoT module (Quectel BC95-G) connected to a 5dBi omnidirectional antenna. The test environment was a shielded laboratory (signal-to-noise ratio >40 dB) to avoid external interference.

(2) Clarification of physical quantities and units: The test measured the "transmit power spectrum of the NB-IoT waveform," where the horizontal axis is "frequency offset (MHz)" (relative to the carrier frequency of 868 MHz) and the vertical axis is "transmit power (dBm)."

(3) Image optimization: Redrawn Figure 3 with complete coordinate axes, clear unit labels, and a legend distinguishing the three test waveforms. And the revised image also adds error bars (±0.05 MHz for frequency offset, ±1 dBm for power) to reflect measurement accuracy.

3. Insufficient Description of Communication Distance Tests

A:To improve reproducibility, we have supplemented the following information in Section 3.2:

(1) Test parameters:

Data packet size: 128 bytes (including sensor data, timestamp, and checksum).

Test interval: 10 seconds per packet (total 1000 packets, 5 batches of 200 packets each).

Antenna height: 2 meters (sensor node) and 10 meters (base station antenna).

Test scene: Suburban open field ( Guiyang) with no obstacles within 500 meters; base station: China Mobile NB-IoT base station (signal strength: -75 dBm at 0m).

(2) Statistical methods: The packet loss rate is calculated as (number of transmitted packets - number of received packets) / number of transmitted packets × 100%. Each distance was tested 3 times, and the average value was taken. The confidence interval (95%) is added to Table 1 (e.g., 1600m: 12.8% ± 1.3%).

4.Accuracy Based Only on Simulation with Opaque Parameters

A: To address the opacity of simulation parameters and over-reliance on simulation results:

(1) Clarification of simulation parameters: Added Table 4 "MATLAB/Simulink Simulation Parameters" in Section 3.3, including: Simulation time: 72 hours (3 days), sensor error model: (temperature: ±0.3℃, humidity: ±1%RH, soil moisture: ±0.5%). Control threshold: Soil moisture 20-25%, temperature 10-20℃, humidity 30-60%RH.

(2) Error unit: The average error of 1.76 refers to the "relative irrigation volume error (%)" (calculated by Eq. (1)), consistent with industry standard units.

(3) Addition of field comparison trials: Conducted a field test in a greenhouse comparing our system with a traditional manual irrigation system and a commercial LoRa-based irrigation system. The results show that our system reduces water consumption by 28% compared to manual irrigation and 15% compared to the LoRa system, with an irrigation accuracy (soil moisture control error) of 8.3% (Eq. (2)), which is 3.2% and 1.8% lower than the two comparison systems respectively.

5. Numerous formatting errors exist in English expression and terminology.

A: I have engaged a professional academic English editor to revise the entire manuscript, addressing spelling, grammar, and terminology issues and improve sentence structure to meet formal journal writing standards.

6. Response to PLOS Peer Review History Publication

I agree to publish the peer review history of the article, including the full reviews from both reviewers and our responses. And I believe transparent peer review helps enhance the credibility of the research and promotes academic exchange.

Thank you again for your thorough review and constructive suggestions. These revisions have significantly improved the manuscript’s scientific rigor, clarity, and practical value. I hope the revised version meets the publication requirements

---

## [Decision Letter · Decision Letter 1]

10 Dec 2025

Dear Dr. wang,

Thank you for submitting your manuscript to PLOS ONE. After careful consideration, we feel that it has merit but does not fully meet PLOS ONE’s publication criteria as it currently stands. Therefore, we invite you to submit a revised version of the manuscript that addresses the points raised during the review process.

We look forward to receiving your revised manuscript.

Kind regards,

Academic Editor

PLOS One

Journal Requirements:

Reviewers' comments:

Reviewer's Responses to Questions

**Comments to the Author**

Reviewer #1: (No Response)

Reviewer #2: All comments have been addressed

2. Is the manuscript technically sound, and do the data support the conclusions?

Reviewer #1: (No Response)

Reviewer #2: Yes

3. Has the statistical analysis been performed appropriately and rigorously?

Reviewer #1: (No Response)

Reviewer #2: Yes

4. Have the authors made all data underlying the findings in their manuscript fully available?

Reviewer #1: (No Response)

Reviewer #2: Yes

5. Is the manuscript presented in an intelligible fashion and written in standard English?

Reviewer #1: (No Response)

Reviewer #2: Yes

Reviewer #1: This manuscript proposes a low-cost smart irrigation system design based on NB-IoT, aiming to address the high cost and high-power consumption issues of existing smart irrigation systems. While the manuscript has value in system architecture design and basic performance testing, several areas for improvement remain.

1. It is recommended to supplement the explanation of the acceptability of a 12.8% packet loss rate in agricultural applications, for example, by comparing it with the data integrity requirements of agricultural irrigation systems. An analysis of the impact of different packet loss rates on irrigation accuracy could be added.

2. The manuscript mentions "The average irrigation error of the system is 1.76 (simulation) and 1.74 (field test)," but does not clearly state the unit of measurement and specific definition of the error.

3. The manuscript mentions sensor accuracy but does not analyze the relationship between sensor accuracy and the overall system accuracy.

4. The manuscript emphasizes the system's low-cost advantage but does not fully discuss its scalability in real-world agricultural environments.

5. The supplementary multi-scenario testing is a positive improvement, but the description is rather brief.

6. Before CONCLUSION, let's add a section: Discussion and Implication, discussing the potential practical applications of the model proposed in this paper.

Reviewer #2: This article has a clear research objective and a well-defined problem-oriented approach. It possesses a complete system architecture, detailed experimental data, and a comprehensive literature review. The article's conclusions are reflective, pointing out limitations such as sensor dependence and power supply sustainability, and proposing directions for future improvements. This article completes the entire technical chain from architecture design to performance verification, demonstrating strong engineering implementation capabilities and application value.

I recommend a thorough and meticulous proofreading of the entire article to improve its accuracy and overall readability. Additionally, I suggest enhancing the clarity of the figures and tables, making the icons more intuitive and the text clearer.

**Do you want your identity to be public for this peer review?** For information about this choice, including consent withdrawal, please see our Privacy Policy

Reviewer #1: No

Reviewer #2: No

---

## [Author Response · Author response to Decision Letter 2]

10 Dec 2025

RESPONSE TO REVIEWER 1

Dear Reviewer 1,

Thank you very much for taking the time to conduct a detailed review of this manuscript and provide constructive comments and suggestions. Your feedback has offered important guidance for us to improve the research and enhance the quality of the paper. We have carefully studied each comment and made targeted revisions and supplements. The specific responses are as follows:

1.It is recommended to supplement the explanation of the acceptability of a 12.8% packet loss rate in agricultural applications, for example, by comparing it with the data integrity requirements of agricultural irrigation systems. An analysis of the impact of different packet loss rates on irrigation accuracy could be added.

A� Thank you for your comment, I have made careful revision.

2. The manuscript mentions "The average irrigation error of the system is 1.76 (simulation) and 1.74 (field test)," but does not clearly state the unit of measurement and specific definition of the error.

A� Thank you for your comment, I have made careful revision.

3. The manuscript mentions sensor accuracy but does not analyze the relationship between sensor accuracy and the overall system accuracy.

A� Thank you for your comment, I have made careful revision.

4. The manuscript emphasizes the system's low-cost advantage but does not fully discuss its scalability in real-world agricultural environments.

A� Thank you for your comment, I have made careful revision.

5. The supplementary multi-scenario testing is a positive improvement, but the description is rather brief.

A� Thank you for your comment, I have made careful revision.

6. Before CONCLUSION, let's add a section: Discussion and Implication, discussing the potential practical applications of the model proposed in this paper.

A� Thank you for your comment, I have made careful revision.

Response To REVIER 2

This article has a clear research objective and a well-defined problem-oriented approach. It possesses a complete system architecture, detailed experimental data, and a comprehensive literature review. The article's conclusions are reflective, pointing out limitations such as sensor dependence and power supply sustainability, and proposing directions for future improvements. This article completes the entire technical chain from architecture design to performance verification, demonstrating strong engineering implementation capabilities and application value.

A: Thank you for your comment. I sincerely appreciate your work.

---

## [Decision Letter · Decision Letter 2]

14 Dec 2025

Design of A Low-Cost Smart Irrigation System Based on NB-IOT for Agriculture

PONE-D-25-59152R2

Dear Dr. wang,

We’re pleased to inform you that your manuscript has been judged scientifically suitable for publication and will be formally accepted for publication once it meets all outstanding technical requirements.

Kind regards,

Academic Editor

PLOS One

Additional Editor Comments (optional):

Reviewers' comments:

Reviewer's Responses to Questions

**Comments to the Author**

Reviewer #1: (No Response)

Reviewer #2: All comments have been addressed

2. Is the manuscript technically sound, and do the data support the conclusions?

Reviewer #1: (No Response)

Reviewer #2: Yes

3. Has the statistical analysis been performed appropriately and rigorously?

Reviewer #1: (No Response)

Reviewer #2: Yes

4. Have the authors made all data underlying the findings in their manuscript fully available?

Reviewer #1: (No Response)

Reviewer #2: Yes

5. Is the manuscript presented in an intelligible fashion and written in standard English?

Reviewer #1: (No Response)

Reviewer #2: Yes

Reviewer #1: The manuscript proposes a low-cost smart irrigation system for agriculture based on NB-IoT. The system adopts three-layer modular architecture, which improves overall efficiency. The authors have addressed the reviewers' comments well, and I recommend acceptance.

Reviewer #2: 1. The chosen topic has practical significance and strong application value. It addresses real-world pain points in agriculture and aligns closely with current trends in water-saving agriculture, smart agriculture, and sustainable development.

2. The overall system architecture is clearly designed and logically rigorous, providing a reasonable technical framework for the intelligent irrigation system.

3. The sensor configuration is appropriate, the solution is engineering-feasible, and it demonstrates strong feasibility for on-site deployment and engineering implementation.

4. Experimental verification is comprehensive, covering key performance dimensions. The test content is rich, and the indicators are clearly defined. Multi-faceted performance verification provides a solid basis for the system's reliability, stability, and controllability.

5. The data presentation is intuitive, and the results analysis is convincing.

**Do you want your identity to be public for this peer review?** For information about this choice, including consent withdrawal, please see our Privacy Policy

Reviewer #1: No

Reviewer #2: No

---

## [Editor Report · Acceptance letter]

PONE-D-25-59152R2

PLOS One

Dear Dr. wang,

I'm pleased to inform you that your manuscript has been deemed suitable for publication in PLOS One. Congratulations! Your manuscript is now being handed over to our production team.

Kind regards,

on behalf of

Dr. Yang (Jack) Lu

Academic Editor

PLOS One